# **Design Considerations of Artificial Mangrove Embankments for Mitigating Coastal Floods – Adapting to Sea-level Rise and Long-term Subsidence**

Hiroshi Takagi1

<sup>1</sup>School of Environment and Society, Tokyo Institute of Technology, Tokyo, Japan

Correspondence to: H. Takagi (takagi@ide.titech.ac.jp)

# Abstract.

Mangrove plantation belts are expected to act as natural infrastructural buffers against coastal hazards. However, their performance will not endure over time if the platform is not appropriately designed. In fact, despite massive funds dedicated to the rehabilitation of mangrove forests, the long-term survival rates of mangroves are generally

- 10 low. This paper investigates the function of mangrove embankments in attenuating the amplitudes of ocean tides through a coupled numerical model that reproduces shallow-water wave propagations under the progress of soil consolidation. The developed model is capable of simulating tidal propagation over an artificial embankment, which will inevitably change its ground surface elevation with the passage of time because of sea-level rise, land subsidence, vegetation growth and sediment accretion. A parametric analysis demonstrates that high tides could
- 15 be effectively mitigated only if the embankment is appropriately designed to maintain an equilibrium state among these multiple influences over the long term. On the other hand, an embankment designed without considering geomorphological transitions will become submerged under the rising sea level, resulting in no significant effect on tidal damping. Therefore, the artificial mangrove embankment must be carefully designed to function not only during the initial stage of its lifetime but also over time, to avoid system failure in the future.


*Keywords*: Ecological disaster risk reduction (Eco-DRR), mangrove, ocean tide, embankment, subsidence, soil consolidation, sea-level rise, sediment accretion, a coupled numerical model


#### 1. INTRODUCTION

Mangrove forests are expected to act as natural infrastructural buffers against natural hazards. In fact, a number of field surveys conducted after recent major disasters such as the 2004 Indian Ocean Tsunami and the 2013 Typhoon Haiyan have confirmed the mitigation effect of mangroves against tsunamis, storm surges, and

- 5 high waves (e.g., Wolanski et al., 2009; Ellison, 2009; Teh et al., 2009; Narayan et al., 2010; Rasmeemasmuang and Sasaki, 2015; Mikami et al., 2016). Such function of vegetation has also been confirmed by laboratory experiments that imposed tsunami-like bores in a wave flume (Irtem et al., 2009; Iimura and Tanaka, 2012; Strusinska-Correia et al., 2013). Takagi et al. (2016a) demonstrated that the impact of a type of tsunami induced by a sudden dyke failure would be substantially mitigated by planting a mangrove belt in front of the dyke,
- 10 through two mechanisms: (1) a reduction in floodwater velocity and inundation depth and (2) a flow smoothing effect, which reduces strong turbulence. Salt marshes have also been shown to significantly reduce wave loads on coastal dykes (Vuik *et al.*, 2016). Vegetation decreases wave height and run-up height as plant diameter and/or stem density increase (Tang *et al.*, 2017).

By protecting coastal environments against shoreline change, mangroves also increase the shoreline's resilience with regard to recovering from a disturbance (Alongi, 2008). Further advantages associated with mangrove include sediment trapping, flood protection, nutrient recycling, wildlife habitat and nurseries (Primavera and Esteban, 2008). These mangrove advantages are collectively referred to as *ecological resilience* (Gunderson *et al.*, 2002).

Given these advantages, the function of mangroves in Ecosystem-based Disaster Risk Reduction (hereinafter
referred to as Eco-DRR) has drawn attention worldwide. Appropriate management of ecosystems can be harnessed to reduce both disaster risks and climate-related risks (UNEP, 2012; UNEP, 2015; CNRD-PEDRR, 2013). In order to implement Eco-DRR, however, the effectiveness of ecosystems in reducing the impacts of natural hazards needs to be quantitatively and practically evaluated. In fact, despite massive funds dedicated to the rehabilitation of mangrove forests over the last two decades, the long-term survival rates of mangroves are generally low, at 10–20% in the case of the Philippines (Primavera and Esteban, 2008).

The present study investigates the application of mangrove forests as a countermeasure for disaster mitigation particularly in urban areas, which are currently experiencing land subsidence and sea-level rise (SLR). For example, Jakarta, one of the fastest growing megacities in the world, experienced subsidence rates varying from 9.5 to 21.5 cm year<sup>-1</sup> in the period between 2007 and 2009, exacerbating coastal flooding issues (Chaussard

*et al.*, 2013; Takagi *et al.*, 2016b). The Chao Phraya Delta in Thailand has been sinking by 5–15 cm year<sup>-1</sup> and the Mekong Delta by 2 cm year<sup>-1</sup> because of intense groundwater use and/or natural consolidation (Giosan *et al.*,

2014; Takagi *et al.*, 2016c). On the other hand, the vertical accretion of sediment in mangrove forests is relatively slow, at the rate of several mm year<sup>-1</sup> (Krauss *et al.*, 2014; Lovelock *et al.*, 2015). Therefore, the response of a mangrove forest to external influences such as SLR and ground subsidence determines the forest's capacity to maintain itself at the pace necessary to adapt to the changing sea level. It is likely that planting mangroves

- 5 without considering such rapid subsidence will result in the forest's submergence under the sea in a short period of the time. When applying an artificial mangrove forest as an Eco-DRR option for coasts experiencing rapid subsidence, an engineering design method that takes into account subsidence as well as SLR will be required to aid planners and engineers in determining the appropriate project specifications.
- There are relatively few studies on tides propagating through mangroves, compared to the number of studies on wind waves, storm surges and tsunamis, which have been extensively studied over the last several decades. Therefore, the present study evaluates mangrove embankments against tidal propagations in order to quantify the function of mangroves in attenuating tidal amplitudes, considering long-term influences such as vegetation growth, SLR, subsidence and sediment accretion. Mangrove platform designs are also investigated to determine which designs can adapt at the same pace as the changing sea level, to maximise the mangrove forest's long-term
- 15 serviceability as a flood mitigation measure.

#### 2. METHODOLOGY

This chapter first describes the concept of an artificial mangrove embankment. The numerical model developed 20 to simulate tidal propagation through a sinking mangrove forest is then briefly explained. Finally, the computational conditions for a case study are described.

#### 2.1 Concept of Artificial Mangrove Embankment

- Coastal areas, particularly in developing countries, are facing increasing disaster risks due to many emerging issues such as rapid population increases, SLR, land subsidence and poor and/or aging infrastructure. Ecosystem platforms are expected to reduce disaster risks as well as climate-related impacts. Figure 1 illustrates (a) a high tide overflowing a coastal dyke and resulting in extensive inundation and (b) an artificial mangrove forest designed to mitigate flood damage, adopted as an Eco-DRR countermeasure. A front part of the embankment is initially formed in a stable trapezoidal shape by adding soil fill with a slope of 1/5 or less (Scenario 1). In this
- 30 initial stage, the embankment functions in the same manner as a dyke structure because the ground surface elevation is higher than the high tides. However, the platform will immediately start sinking because of soil

consolidation resulting from the soil's weight and/or forced displacement due to land subsidence in adjacent lands. Two additional scenarios that could occur over many years are also illustrated in the figure. Scenario 2 represents the case in which the embankment is maintained with the changing sea level by achieving an equilibrium state between subsidence, SLR and sediment accretion over the mangrove bed. On the other hand,

5 Scenario 3 shows that the embankment designed without considering long-term evolutions will become submerged under mean sea level (MSL). In this state, the mangrove forest cannot expand because high waves pass across the embankment without sufficient attenuation, resulting in less sediment trapping and no significant effect on tidal damping.

### 10 2.2 Shallow Water Wave - Soil Consolidation Integrated Model

Waves propagating in shallow water (relative water depth: h/L < 1/20) are called shallow water waves (or long waves); this wave type includes ocean tides, tsunamis and storm surges. Shallow water waves can be simulated by solving the following governing equations: the equation of motion, Equation (1), and the continuity equation, Equation (2) (e.g., Kowalik and Murty, 1993):

15

$$\frac{\partial U}{\partial t} + U \frac{\partial U}{\partial x} + g \frac{\partial \eta}{\partial x} + \frac{g n^2 |U| U}{h^{\frac{4}{3}}} = 0$$
(1)

$$\frac{\partial \eta}{\partial t} + \frac{\partial (h+\eta)U}{\partial x} = 0$$
<sup>(2)</sup>

In Equations (1) and (2), U is the depth-averaged velocity,  $\eta$  is the water displacement, h is the water depth, g is 20 the gravitational acceleration and n is Manning's n value. In the present model, these governing equations are discretised by the finite difference method with staggered grids. The spatial derivatives are approximated by central differences, whereas the time derivatives are discretised by forward differences.

Consolidation is the process whereby an increase in stress gradually causes a volume reduction as water is released from the soil. This term should not be confused with compaction, which is an instantaneous change in

25 the soil volume due to a mechanical impact. In the shallow water wave – soil consolidation integrated model developed for this study, the water depth used in Equations (1) and (2) is adjusted to the ground subsidence that is calculated by the following Terzaghi one-dimensional consolidation equation (e.g., Wesley, 2010; Knappett and Craig, 2011):

where  $C_v$  is the coefficient of consolidation,  $u_e$  is the excess pore pressure during consolidation, t is time and z is the vertical ordinate.

5 A non-dimensional form of Equation (3) is derived by introducing  $Z = z/H_d$  and  $T = C_v t/H_d^2$ :

$$\frac{\partial u_e}{\partial T} = \frac{\partial^2 u_e}{\partial Z^2} \qquad , \qquad (4)$$

where  $H_d$  is the drainage path length, which is either half the layer thickness (when drainage can occur at both boundaries) or the full layer thickness (when drainage can only occur at one boundary). Equation (4) is equivalent to the 1-D heat equation and can be solved analytically with initial and boundary conditions. The solution is derived by assuming the initial condition of  $u_e$  as a load  $q_s$  created by the embankment as follows:

$$u_e = q_s \sum_{n=0}^{\infty} \frac{2^{(-1)^n \cos(a_n Z)}}{a_n} \exp(-a_n^2 T) \qquad , \qquad (5)$$

15

where  $a_n = (2n + 1)\pi/2$ . The estimated final subsidence is then calculated by Equation (6) with a compression index  $C_c$  that is assumed to remain constant throughout a compressible soil. The subsidence  $S_c$  at a given time Tcan also be calculated by Equations (7) and (8) with the degree of consolidation U.

20

$$S_f = \frac{C_c}{1+e_0} H \log_{10} \left( 1 + \frac{q_s}{\sigma'} \right) \tag{6}$$

$$S_c = S_f U(T) \tag{7}$$

$$U(T) = 1 - \sum_{n=0}^{\infty} \frac{2}{a_n^2} \exp(-a_n^2 T)$$
(8)

The effective vertical stress  $\sigma'$  in Equation (6) can be calculated by referencing the method of vertical stress 25 due to embankment loading (Osterberg, 1957).

Although the present paper considers only the subsidence  $S_c$  associated with soil consolidation due to the surplus weight of the embankment, regional land subsidence  $S_r$  (often caused by groundwater withdrawals) should also be considered in relevant cases, leading to the following resultant subsidence  $S_t$ .

# $S_t = S_c + S_r \tag{9}$

### 2.3 Resistance of Mangrove Forest

The tidal flows become more complex when eddies, jets and stagnation are created by the resistance of the mangrove forest. Vegetation resistance is composed of two factors: energy losses due to friction and drag due to

- the variable velocities (Furukawa and Wolanski, 1996; Lewis, 1997; Green, 2005). Drag force related to plant length, diameter, Reynolds number and vegetation density have been simulated by previous studies (e.g., Mazda *et al.*, 1997; Strusinska-Correia *et al.*, 2013; Maza *et al.*, 2015; Losada *et al.*, 2016). Because the shallow water wave model represented by Equation (1) accounts for only bottom friction, the effect of drag can be taken into account solely through Manning's *n* value. Whereas coastal or riverine areas without dense vegetation are often
- assumed to have a relatively small *n* value in the range of 0.02–0.03 (Bricker *et al.*, 2015), a much greater value needs to be applied when the flow in an area of dense vegetation is simulated. In addition, coefficient values tested on large-depth flows should not be simply applied to small-depth flows such as a vegetation bed (Diaz, 2005). Table 1 summarises the Manning's *n* values that were proposed in the previous papers on flood mitigation effects of various type of vegetation. Although the *n* values vary according to conditions, the present study
- assumes the *n* value of 0.3 (s m<sup>-1/3</sup>) in Equation (1) to represent a fully grown mangrove forest; this value was selected as the average of those listed in Table 1. On the other hand, the value for a bare embankment condition (namely, landfill with no mangrove vegetation) was assumed to have a much smaller *n* value of 0.025, which is commonly used for the roughness of a seabed (Bricker *et al.*, 2015).

#### 25 2.4 Sediment Accretion Rate

Mangroves are expected to naturally respond to relative sea-level rise and thus may not require maintenance as a flood protection measure. Accretion in these wetlands occurs because of the accumulation of organic matter produced by the plants themselves (Valiela, 2006). Many field investigations on sediment trapping rates suggest that many coastal marshes accrete at rates comparable to SLR (e.g., Lynch *et al.*, 1989; Day and Templet, 1989;

Wood *et al.*, 1989; Kearney *et al.*, 1994; Parkinson *et al.*, 1994). For instance, Blum and Christian (2004) present an organic sediment accretion of 9 mm yr<sup>-1</sup> for a marsh covered by *Spartina alterniflora* in Virginia, in the United

States. Saad *et al.* (1999) conducted an extensive two-year field survey in an estuarine mangrove swamp. They sampled surface sediment in both non-monsoon and monsoon periods at 52 stations and found that the accretion rate was  $1.46 \pm 0.13$  cm yr<sup>-1</sup> and  $0.66 \pm 0.04$  cm yr<sup>-1</sup> for the first and second year, respectively, for an average of 1.06 cm yr<sup>-1</sup>. Although the annual deposit rate varies to a great extent, based on these previous studies, the present

study assumes that sediment accretion will occur at the rate of 1 cm  $yr^{-1}$  over the embankment's long-term transition induced by the amount of time over which the embankment's surface is lower than MSL.

# 2.5 Computational Conditions for the Case Study

A case analysis was conducted to investigate how mangrove plantations can mitigate tide propagation across

- the embankment. Numerical settings and other important assumptions applied in the numerical analysis are summarised as follows.
  - Time span of the analysis: 100 years after the completion of the landfill.
  - Spatial grid: 50 m  $\times$  600 grids = 30 km in the cross-sectional direction.
- Original bathymetry: Varying from a 10-m depth offshore to a 1.5-m depth onshore;
  - the mangrove embankment is created on top of the original bathymetry.
  - Embankment heights (thicknesses): 2, 3, 4 and 5 m.
  - Time increment for shallow water wave computation: 0.2 s.
  - Boundary condition: A diurnal tide with amplitude of 0.3 m.
- Computational time: 24 hours, which can fully encompass one tidal cycle.
  - Sediment accretion on the embankment will start after the mangrove plants are fully grown.
  - Mangrove plants will be fully grown in 10 years.
  - Bottom roughness, represented by Manning's n, will linearly increase from the initial value of 0.025 (the embankment not covered by vegetation) to the final value of 0.30 (fully covered by vegetation) over the
- growing period.
  - Soil consolidation parameters:  $C_c = 1.0$ ;  $C_v = 85.0$  cm<sup>2</sup> day<sup>-1</sup>.
  - A mangrove embankment filled by sand (unit weight of 21 kN m<sup>-3</sup>) is to be placed on top of the clay seabed layer (unit weight of 18 kN m<sup>-3</sup>); the consolidation process occurs only in the clay layer, of which the thickness is assumed to be 18.5 m (from -1.5 m down to -20 m MSL).

Because there are many parameters that influence the scenarios in the case analysis, several parameters such as soil parameters and tidal condition were determined by reference to the condition in the Jakarta Bay (Takagi *et al.* 2016b).


# **3. RESULTS**

In this section, the performance of the mangrove embankment in mitigating tidal propagation is discussed based on the results of the case analysis.

#### 3.1 Changes in Ground Elevation over Many Years

- Figure 2 (b) shows the hypothetical geometry of the mangrove embankment created by filling soil on a shallow water bay with a constant water depth of 1.5 m, as illustrated in Figure 2 (a). Here, two initial design selections are considered as examples: a high embankment with a mound thickness of 5 m (elevation +3.5 m) and a low embankment with a mound thickness of 2 m (+0.5 m). Although both of these embankments are expected to become submerged after 60 years as shown in Figure 2 (c), the ground surface remains close to the sea level
- 15 when the initial thickness is set to be 5 m. On the other hand, for the case of a 2 m thickness, the water depth was found to be nearly 1.5 m, which is the same as the pre-fill soil level. Therefore, a high-elevation embankment may be the countermeasure with the best long-term performance, whereas a low-elevation embankment may function effectively in the very beginning, but would lose functionality after many years.
- Because mangroves function to trap sediment, mangrove forests could become an important sink for suspended sediment (e.g., Woodroffe, 1992; Furukawa *et al.*, 1997). Given the situation in which both SLR and subsidence inevitably occur, sediment accretion only raises the ground level and keeps the mangrove surface even with the sea level. Therefore, it would not be possible to achieve a sustainable mangrove embankment system in the case that organic or other suspended matter fails to sink to the ground surface, e.g., due to strong turbulence and sediment pick-up caused by high waves.
- Figure 3 illustrates how changes in the ground elevation over time could differ with and without sediment accumulation. The difference is particularly clear for the case of the 5-m-high embankment, shown by the blue continuous line. In the first 40 years, sediment accretion is not yet calculated because the ground elevation remains above the water surface; hence, the identical lines for the cases with and without sediment accumulation over this time period. However, at approximately the 50<sup>th</sup> year, the ground starts to become inundated. The
- 30 embankment with accretion, which receives sediment deposition at the rate of 1 cm  $yr^{-1}$ , is likely to maintain its surface level with the sea level, demonstrating a successful tide attenuation scenario. In contrast, the lower


embankment, such as that with a 2-m-height, may not substantially contribute to a reduced water level. This is because the water depth above the ground is too deep to effectively reduce the tidal energy, irrespective of whether sediment accretion is expected.

# 5 3.2 Tidal Propagations over the Mangrove Embankments

Suspended sediment trapped by mangroves is highly dependent on the extent to which the plants are fully grown. The water depth above the embankment and the embankment width appear to be predominant influences on tidal attenuation. Among many influential factors, these three parameters in particular must be carefully examined in the designing phase. Figure 4 shows how the effectiveness of the mangrove embankment as a tidal attenuator would vary with changes in those values.

The present model assumes that the vegetation growth ratio starts from 0%, and the vegetation growth ratio is assumed to linearly increase up to 100% after 10 years. The tidal amplitude is not significantly attenuated when the vegetation growth ratio is low. For example, tidal propagation is not disturbed when the water depth is 1 m and no vegetation covers the ground. However, if the mangrove embankment can be designed to be as wide as

500 m, an amplitude reduction of approximately 40% is expected when the plants are fully grown. The figures demonstrate that the embankment width can be shortened by 200 m, with a corresponding tidal amplitude reduction 40–50 % if the water depth can be kept lower than 50 cm.

In addition, the backwash during the ebb tide is hindered because of the bed friction increased by the vegetation. The friction increase also induces a delay in the tidal phase of up to a few hours. This retention of

- 20 seawater should provide a favourably serene water environment for marine organisms, and the currents around the vegetation create stagnation zones where the suspended sediment will settle. On the other hand, water pollution or flood issues may also arise in highly populated coasts because of the inevitable water stagnation. The time series demonstrates that the tide becomes asymmetrical in mangroves because of the resistance offered by mangrove vegetation. The signal modification is clearly apparent starting at the time when the water begins to
- 25 inundate the swamp, and it is especially pronounced around the time when the bottom substrate becomes dry at ebb tide, as presented by Mazda and Kamiyama (2007).

Figure 5 summarises the tidal attenuation ratios that indicate to what extent an incident tide is attenuated by the mangrove embankment according to the embankment's width and the water's depth. The maximum attenuation at the water depth of 30 cm and the width of 500 m is 64% when the vegetation is fully grown,

whereas only 12% attenuation is expected immediately after the construction of the embankment. Equation (1) indicates that the bottom friction rapidly increases in proportion to the square of the velocity. Therefore, the

growth ratio, which indicates how strongly the vegetation resists the tidal flow, appears to be one of the predominant design factors that should be considered.



### 4. DISCUSSION

The results presented in the previous section confirm that the mangrove embankment will not effectively mitigate tidal floods unless it is carefully designed with forethought regarding future morphological transitions. On the other hand, it may be possible to achieve an equilibrium state if the elevation of the mangrove embankment is balanced between multiple factors such as sea-level rise, land subsidence, vegetation growth and sediment accretion. In this case, the system is expected to be sufficiently sustainable that expensive maintenance

work will not be required over the mangrove forest's lifetime. The following insights are expected to help planners design effective mangrove embankments.

Initial elevations must be sufficiently high to avoid being totally submerged: Mangroves can cover a wide intertidal area, which is flooded at high tide and exposed at low tide, and mangroves normally grow in areas between the middle tide level and the highest high water spring tide. Within the intertidal zone, species prefer different elevations, salinities and inundation frequencies (Global Nature Fund, 2007; Wolanski *et al.*, 2009). Therefore, the mangrove plantation is expected to be most effective within the intertidal zone. However, this general guideline may not always be successful when the plantation is implemented on sinking ground.




**Narrower mangrove embankments will not be effective as tidal attenuators:** Unlike wind waves, long tidal waves cannot be easily attenuated. A wider embankment covered by mangroves is thus required for a substantial mitigation effect. However, on most urban coasts with high populations, mangrove plantations may be allowed within only limited spaces. In such a case, the effectiveness of a narrow embankment should not be overestimated as a tidal attenuator.

**Mangroves retain seawater, creating both positive and negative influences:** Increased resistance due to the high density of mangrove vegetation induces the retention of seawater on the ground during the ebb tide phase. A tranquil water environment is beneficial for marine organisms and animals and thereby fosters biodiversity; on the other hand, stagnant water may exacerbate urban floods because the stagnant water may inhibit the discharge