# Peer review of "Design Considerations of Artificial Mangrove Embankments for Mitigating Coastal Floods – Adapting to Sea-level Rise and Long-term Subsidence"

_Natural Hazards and Earth System Sciences, 2017_

## Referee Comment (RC1) · Anonymous Referee #1 · 31 Mar 2017

*General comments:*

This paper investigates the benefits of mangrove embankments in attenuating the amplitudes of ocean tides through a coupled numerical model that reproduces shallow-water current propagations under the effect of soil consolidation and sea level rise. This phenomenon/process is a typical water-sediment-topography interaction issue can be described as below:

[Figure]

Ocean waves/tides induced onshore/offshore current causes accordingly sediment transport due to the bottom shear stress, then the net balance of sediment flux result in erosion or deposition of movable layer and further cause topographical change. Finally, current is influenced by the renewed topography. Basically, it is a dynamic cycle and is significant in the nearshore area. Thus in my opinion, the method used to investigate this topic should be capable of reflecting the interaction processes aforementioned. However the developed model in this study seems only cover part of the cycle mentioned above, sediment transport due to the shear stress generated by current is not considered. Topographical change calculation only consider soil consolidation (Eq.3 ~ 8), lack of erosion/deposition caused by net sediment flux. The sediment accretion(deposition)/decrement(erosion) should be calculated according to the temporal of vegetation growth ratio not only an assumed average accretion rate 1 cm/year instead. Generally speaking, the topic discussed in this paper is essential and well fit with the scope of this journal. However, the completeness of the coupled model applied in this study, in my opinion is insufficient as well as lack of model verification based on field data (hydrodynamic, topography etc.). In additions, it is suggested to extend more discussion on the sand trapping efficiency and tidal wave energy dissipation due to the existence of mangrove embankments especially in different stages.

***Specific comments:***

1. P.3, I suggest to add a paragraph to introduce the study area. For example climate, shore geomorphology, wave, tidal components, historic disasters etc. That will be helpful for readers to understand the determination of computational conditions for the case study (P.7).

2. P.4, Ln26-27, "...the water depth used in Equations (1) and (2) is adjusted to the ground subsidence that is calculated by the following Terzaghi one-dimensional consolidation equation...", How to adjust need more descriptions.

3. P.4, please provide field verification examples or application references of this integrated model.

4. P.7, 2.5 Computational conditions for the case study:
   - What about the sea level rise scenario during simulation period?
   - Why boundary condition only consider diurnal tide? What about other tidal components?
   - Bottom roughness linearly increase with time to represent vegetation growing, however the sediment accretion was assumed a constant value 1 cm/year seemed not reasonable. Sediment accretion could be higher in the final stage when the land subsidence become worse.

5. P.9, Ln6, "Suspended sediment trapped by mangroves is highly dependent on the extent to which the plants are fully grown", this means the amount of trapped sediment should be calculated according to the shear stress generated by current.

6. P.9, Ln18-26, P.10, Ln1-2, P.10, Ln27-30, please provide more current simulation results like velocity, shear stress variation etc.

7. P.11, Ln4-9, "...that erosion of the embankment would decrease because hydraulic energy is dissipated by the vegetation. Therefore, these considerations relating to how the sediment accretion could be promoted may be critical to successful mangrove embankment designs...", it is suggested to further discuss the sediment trapping efficiency in different stages.

8. P.18, Figure 4, it is suggested to move "growth ratio --%" to a proper location since it will confuse with the title of vertical axis (water elevation).

9. P.19, Figure 5, What is the definition of "Attenuation ratio (%)"?

---

## Referee Comment (RC2) · Anonymous Referee #2 · 29 May 2017

May 2017-05-28 Review Nat Hazards

General comments

The paper addresses a topical theme and provides important insight for the design of mangrove restoration. Mangrove restoration has sparked in popularity in an attempt to make up for their alarming global loss and because the features they offer for coastal protection is particularly interesting for adaptation and risk reduction. However, successful restoration cases have been, in general, very scarce. Rates of successes have been characteristically low, in part due to lack of adequate designs for initial protection of seedlings and embankments. This paper specifically addresses the latter point and provides useful information for restoration on the ground. I suggest its publication after

addressing a few key points.

First, the paper focuses on tide attenuation to show the potential of mangroves to reduce coastal flooding. However, it is not clear from the onset of the paper that 'mitigation of coastal floods' only refers to 'ocean tides'. Furthermore, if it is unclear if this term refers only to astronomical tides throughout the paper since other drivers of flooding are cited and discussed. The paper refers to waves, storm surges, tides and tsunamis. While wind waves are short waves; surges, tsunamis and (astronomical) tides are long-waves that behave differently to each other. This point must be clarified and the author should provide a clear definition of what tidal propagation refers to within the scope of the paper. In the same regard, scope and goal of the paper could be better framed in Figure 1 too (see specific comments below).

Similarly, it is confusing how deterministic astronomical tides are relevant in the context of EcoDRR and catastrophic coastal flooding. Tides are one of the factors of extreme sea levels but they rarely are the main driver in extreme flooding events. It is assumed that tide propagation is used here as way to measure flood reduction more broadly (storm surges?) and given that the equation is the same for both types of long waves, but this requires further clarification.

On the same token, it should be clarified that tide attenuation is only tested for specific tide characteristics (1 amplitude and 1 tide period in Figure 4), for different geometries of the embankment. While the analyses focuses on highlighting the effects on the different embankment geometries, the discussion should merit some acknowledgement of the fact that flood events with different characteristics (amplitude and celerity-duration) would behave differently.

Also, some reorganization and text improvement should help to facilitate the reading and an overall understanding of the paper main contributions. Some suggestions are included in the specific comments.

Finally, it should be clarified (maybe in discussion) that the paper refers only to tide

attenuation and the consolidation of the embankment over time but other factors, such as the diversity of species, are also critical for flood reduction and good ecological performance. Diverse species increase friction (manning coefficient) in the mangrove forests and reduces flooding but are also a key component to becoming functional ecosystems, in the context of Eco-DRR. Many restoration cases in the past have failed to recognize the species diversity as a clear factor in restoring mangrove forests for flood protection.

Specific comments

Page 2 line 8-10 – Phrase unclear. line 10-12 – rephrase. Maybe: mangroves provide key ecosystem services such as: . . . Also, explain ecological resilience in this context.

line 6. Move 'salt marshes have also . . .' after 'vegetation decreases. . . '

line 21. Not all urban areas are suffering subsidence and sea level rise. Clarify and express better that: 'mangroves can help reduce risk in urban areas under threat from sea level rise and subsidence'

line 29-31. See (Sasmito et al. 2015) for a discussion on mangroves and sea level rise.

Page 3 Line 3 onwards. Clarify and define tides, surges and tsunamis. Line 13. Remove or describe sections at the end of the introduction Line 25-27. How is it affected by erosion of the waterfront? Clarify that erosion of the waterfront (for example from wave action) is not considered in the model.

Line 29 onwards. While figure 1 describes the overall approach and design of the numerical experiment, the text in section 2.1 is described as conclusions rather than methods or hypothesis that the paper will test. It is suggested to rewrite this paragraph to better express what scenarios the experiment will address and compare.

Line 6. Describe differences between surges, tides and tsunamis and that only tides will be used in the paper.

[Figure]

Page 5. Where is ue (eq 5) used in eq 6 and 7?

Page 5 and 6. It is unclear where sediment deposition (later analyzed in the paper) appears in these equations. Is this through a linear sediment accretion rate (section 2.4)?

Line 31. MSL : Mean Sea Level

Page 7. Section 2.5. Format or describe the case study.

Line 30. Consider removing.

Page 8. 1st paragraph. Only by soil consolidation?

Line 12. Because mangroves trap sediment... Line 24. Tide attenuation scenario is unclear. Line 26. Reduce the tidal energy? Unclear.

Page 9. Line 1-3. Outline better the 3 factors: plants height, water depth, embankment width and depth.

Line 4. Define vegetation growth rations. This variable is also used in subsequent figures but is unclear what it refers to.

Line 12-15. This paragraph is discussion material.

Paragraph in line 20. How tidal amplitude influences the analysis? It is not defined in the paper that tide attenuation is only tested for a specific tidal amplitude and tide period. However, flood attenuation would depend on the wave amplitude and celerity (i.e. period) in relation to the geometry of the embankment (depth and width), as seen in equations. In other words, for the same tide properties, different embankments geometries provide different attenuation factors, but for the same embankments, different tides would exhibit different behavior too. If sensitivity of the results to tides (and other flood events) is not explored, it should be properly acknowledged and described.

Line 30. The results presented confirm . . .

Page 10. Line 2. Use SLR consistently throughout Line 3. Tide amplitude and period too?

Line 19. Both potential and negative effects: Line 19. Paragraph also discussed in results. Consider consolidating.

Line 29. Erosion of embankments usually occurs from wave action at the forest front when the area is not properly sheltered from it.

Line 30. Diversity of species in the forest should also be consider to provide better frictional drag and functional ecological performance.

Page 11. Define and use MSL for mean sea level consistently throughout Line 4. Use static rather than stable. Line 6. Rewrite. For example: For example, climate patterns such as El Niño Southern Oscillation can increase sea level and these variations have been shown to larger than historic trends in sea level rise (e.g. Losada et al. 2013).

The authors may consider adding a final discussion point on how these results could inform flood attenuation for other events that are not ocean tides.

Line 16. Keep the mangrove surface level up with . . .

Line 20. These values of attenuation are specific to the tide amplitude and period.

Technical comments on Figures

Figure 1. It is confusing that ocean tides are here related to catastrophic flooding on urban areas (e.g. panel a). The sketch and text in the corresponding section should be more clear on what the sketches represent and what is the particular scope of the paper.

Figure 3. add the sedimentation rate. 1cm/yr?

Figure 4. Define "vegetation growth rates" and "growth ratio". Do they represent the same variable?

Figure 5. The 3D perspective is confusing. Suggestion: 3 panels, one for each water depth.

ADDITIONAL REFERENCES

Losada, I.J., Reguero, B.G., Méndez, F.J., Castanedo, S., Abascal, A.J. & Mínguez, R. (2013). Long-term changes in sea-level components in Latin America and the Caribbean. Glob. Planet. Change, 104, 34–50. Sasmito, S.D., Murdiyarso, D., Friess, D.A. & Kurnianto, S. (2015). Can mangroves keep pace with contemporary sea level rise? A global data review. Wetl. Ecol. Manag., 24, 263–278.
* * *

---

## Author Comment (AC1) · 7 Jul 2017

The comment was uploaded in the form of a supplement:
https://www.nat-hazards-earth-syst-sci-discuss.net/nhess-2017-61/nhess-2017-61-AC1-supplement.pdf

---

## Author Comment (AC2) · 7 Jul 2017

Dear Editor and Reviewers, NHESS,

First of all, we deeply appreciate the editor and reviewers' efforts to evaluate our manuscript, and also must thank them for the fact that they spent their precious time in conducting this reviewing process. The authors wish to express their gratitude for a number of constructive comments and advice given regarding the original manuscript, which much assisted the authors in its revision. Please find below a detailed reply to all reviewers' comments.

Kind Regards,

Hiroshi Takagi
Tokyo Institute of Technology

Reply to the comments:
In blue: reviewers' comments
In red: authors' reply

**Reviewer #1**

General Comments and Remarks

This paper investigates the benefits of mangrove embankments in attenuating the amplitudes of ocean tides through a coupled numerical model that reproduces shallow-water current propagations under the effect of soil consolidation and sea level rise. This phenomenon/process is a typical water-sediment-topography interaction issue can be described as below:

[Figure]

Ocean waves/tides induced onshore/offshore current causes accordingly sediment transport due to the bottom shear stress, then the net balance of sediment flux result in erosion or deposition of movable layer and further cause topographical change. Finally, current is influenced by the renewed topography. Basically, it is a dynamic cycle and is significant in the nearshore area. Thus in my opinion, the method used to investigate this topic should be capable of reflecting the interaction processes aforementioned. However the developed model in this study seems only cover part of the cycle mentioned above, sediment transport due to the shear stress generated by current is not considered. Topographical change calculation only consider soil consolidation (Eq.3 ~ 8), lack of erosion/deposition caused by net sediment flux. The sediment accretion(deposition)/decrement(erosion) should be calculated according to the temporal of vegetation growth ratio not only an assumed average accretion rate 1 cm/year instead. Generally speaking, the topic discussed in this paper is essential and well fit with the scope of this journal. However, the completeness of the coupled model applied in this study, in my opinion is insufficient as well as lack of model verification based on field data (hydrodynamic, topography etc.). In additions, it is suggested to extend more discussion on the sand trapping efficiency and tidal wave energy dissipation due to the existence of mangrove embankments especially in different stages.

We thank the reviewer for the great number of very productive comments and suggestions above, which would enable us to significantly improve our manuscript.

We agree the importance of the dynamic interactions among current, sediment, and topographical changes (here, referred to as short-term mechanisms), as illustrated by the reviewer's diagram. Indeed, we didn't state the contribution of the short-term mechanisms on the morphological changes of the mangrove embankment. Therefore, a brief section "Short-term morphology changes under waves and tidal currents" will be added in the revised manuscript, stating "These appear to be very important mechanisms in determining the topography of the mangrove embankment in the short term. Nevertheless, a beach profile due to this mechanism will be seasonally fluctuating around an equilibrium beach profile if it is monitored over many years. This cross-shore evolution is believed to occur on a somewhat shorter time scale than the planform evolution, and tends to demonstrate a seasonal cycle (i.e. summer waves act like bulldozers that push the offshore sand up across the shoreline, whereas the winter waves rush onto the beach, erode the beach sand, and carry it seawards with the backwash). Because this study aims at investigating the long-term / macroscopic evolution of the

embankment, the morphology changes under waves and/or tidal currents were not taken into account in this study; it focused only on the cycles of the long-term mechanisms."

Besides, a conceptualized figure (borrowing the idea of the reviewer's diagram) will be added, as shown below, so that the scope and the applicability of the proposed model could be clearly understood by the readers.

[Figure]

**New figure to be inserted in the revised manuscript**: Scope and the applicability of the proposed model. Long and short-term mechanisms could both cause morphology changes of a mangrove embankment. However, only the long-term mechanisms were taken into consideration in this study. The shallow-water wave model adopted here can reproduce the propagation of not only tides but also storm surges and tsunamis over the embankment. However, the latter mechanisms may not be always reproduced.

Specific comments:
1. P.3, I suggest to add a paragraph to introduce the study area. For example, climate, shore geomorphology, wave, tidal components, historic disasters etc. That will be helpful for readers to understand the determination of computational conditions for the case study (P.7).
We agree. The section of "Computational Conditions for the Case Study" will be modified by including some more clarifications on the study area.

2. P.4, Ln26-27, "… the water depth used in Equations (1) and (2) is adjusted to the ground subsidence that is calculated by the following Terzaghi one-dimentional consolidation equation…", How to adjust need more descriptions.

We agree. The following paragraph and some more will be added in the revised manuscript:

"The water depth used in the shallow-water wave model will be updated by incorporating the calculated subsidence at the $n$th year. Besides, the SLR and sediment accretion will be taken into account in determining the water depths above the ground surface of the mangrove embankment."

3. P.4, please provide field verification examples or application references of this integrated model.

The shallow-water wave model and the Terzaghi one-dimensional consolidation theory are both very commonly used in scientific communities with the long-history of engineering application. Moreover, the present coupled model do not consider the short-term mechanisms as mentioned earlier. Also, the verification of the model seems to be virtually impossible as it requires the data measured over decades. On the other hand, the verification could be possible for the tidal propagation part if the field observed data could be obtained in a natural or manmade mangrove forest, of which geometry is similar with one assumed in this study. However, such model verifications are outside the scope of the present paper, and should be the target of future work.

4. P.7, 2.5 Computational conditions for the case study:
- What about the sea level rise scenario during simulation period
The clarification on the SLR scenario will be added in the revised manuscript.

- Why boundary condition only consider diurnal tide? What about other tidal components?

In this study, the analysis was performed only for the diurnal tide to simplify the discussion and to test for an unfavourable tidal condition, where the mangrove's effect as a tidal attenuator could be limited. The longer the tidal period, the smaller the tide attenuates. Therefore, the authors believe that investigating the diurnal tide is ideal for observing the fundamental function of the mangrove embankment system. The tidal damping for semi-diurnal tidal components, which are half the periods of the diurnal tides, appears to be more remarkable compared to the diurnal tide, when their amplitudes are the same.

- Bottom roughness linearly increase with time to represent vegetation growing, however the sediment accretion was assumed a constant value 1 cm/year seemed not reasonable. Sediment accretion could be higher in the final stage when the land subsidence become worse.

5. P.9, Ln6, "Suspended sediment trapped by mangroves is highly dependent on the extent to which the plants are fully grown", this means the amount of trapped sediment should be calculated according to the shear stress generated by current.

Thank you for providing this insightful comment. Indeed, it is expected that a fully-grown mangrove could substantially slow down tidal velocity, thereby contributing to accelerated sediment deposit. In this study, however, the rate of sediment accretion was simply assumed to be constant because the uncertainty inherent in the process is too large to provide a reliable estimation over the years. It is also noted that the efficiency of sediment trap with vegetation growth could be canceled by the velocity increase due to the water-depth increase as the land subsides.

6. P.9, Ln18-26, P.10, Ln1-1, P.10, Ln27-30, please provide more current simulation results like velocity, shear stress variation etc.

We agree. The figure dealing with the flow velocity over the mangrove embankment, such as below, will be place and discussed in the revised manuscript.

[Figure]

**New figure to be inserted in the revised manuscript**: Cross-sectional profiles of flow velocity during one tidal cycle (line was drawn every 6 hours). The water depth of the embankment was assumed 50 cm.

---

## Author Comment (AC3) · 7 Jul 2017

Dear Editor and Reviewers, NHESS,

First of all, we deeply appreciate the editor and reviewers' efforts to evaluate our manuscript, and also must thank them for the fact that they spent their precious time in conducting this reviewing process. The authors wish to express their gratitude for a number of constructive comments and advice given regarding the original manuscript, which much assisted the authors in its revision. Please find below a detailed reply to all reviewers' comments.

Kind Regards,

Hiroshi Takagi
Tokyo Institute of Technology

Reply to the comments:
In blue: reviewers' comments
In red: authors' reply

**Reviewer #2**

General comments
The paper addresses a topical theme and provides important insight for the design of mangrove restoration. Mangrove restoration has sparked in popularity in an attempt to make up for their alarming global loss and because the features they offer for coastal protection is particularly interesting for adaptation and risk reduction. However, successful restoration cases have been, in general, very scarce. Rates of successes have been characteristically low, in part due to lack of adequate designs for initial protection of seedlings and embankments. This paper specifically addresses the latter point and provides useful information for restoration on the ground. I suggest its publication after addressing a few key points.
We thank the reviewer for the great number of very productive comments and suggestions, which would enable us to significantly improve our manuscript.

First, the paper focuses on tide attenuation to show the potential of mangroves to reduce coastal flooding. However, it is not clear from the onset of the paper that 'mitigation of coastal floods' only refers to 'ocean tides'. Furthermore, if it is unclear if this term

refs only to astronomical tides throughout the paper since other drivers of flooding are cited and discussed. The paper refers to waves, storm surges, tides and tsunamis. While wind waves are short waves; surges, tsunamis and (astronomical) tides are long-waves that behave differently to each other. This point must be clarified and the author should provide a clear definition of what tidal propagation refers to within the scope of the paper. In the same regard, scope and goal of the paper could be better framed in Figure 1 too (see specific comments below).

Sorry for the insufficient explanation on the types of coastal flood. Although astronomical tides are only discussed in this paper, mangrove is expected to mitigate storm surges, tsunamis, and high waves in addition to tides. Due to the theoretical limitation of shallow-water-wave model, tidal flood is mainly addressed. The relative depth, which is defined as the water depth divided by the wave length, is commonly used to classify the types of waves. If the relative depth is smaller than 1/20, the given wave can be considered as a shallow-water wave. Given this criteria, not only tides but also storm surges and tsunamis are considered shallow-water waves. Nevertheless, abrupt changes in the water levels may also appear during tsunamis or storm surges as a form of bore or soliton fission. Because mangrove mainly inhabits very shallow-depth regions, these strong non-linear characteristics will be particularly pronounced. The depth-averaged velocity assumption adopted in the equations may not be applicable if such a mechanism occurs. On the other hand, it is obvious that ordinal wind waves of periods less than 20 s are by no means shallow-water waves. Therefore, it is not appropriate to apply the model to wind waves. In this way, the scope and the applicability of the proposed model will be clarified in the revised manuscript.

[Figure]

**New figure to be inserted in the revised manuscript**: Scope and the applicability of the proposed model. Long and short-term mechanisms could both cause morphology changes of a mangrove embankment. However, only the long-term mechanisms were taken into consideration in this study. The shallow-water wave model adopted here can reproduce the propagation of not only tides but also storm surges and tsunamis over the embankment. However, the latter mechanisms may not be always reproduced.

Similarly, it is confusing how deterministic astronomical tides are relevant in the context of EcoDRR and catastrophic coastal flooding. Tides are one of the factors of extreme sea levels but they rarely are the main driver in extreme flooding events. It is assumed that tide propagation is used here as way to measure flood reduction more broadly (storm surges?) and given that the equation is the same for both types of long waves, but this requires further clarification.

On the same token, it should be clarified that tide attenuation is only tested for specific tide characteristics (1 amplitude and 1 tide period in Figure 4), for different geometries of the embankment. While the analyses focuses on highlighting the effects on the different embankment geometries, the discussion should merit some acknowledgement of the fact that flood events with different characteristics (amplitude and celerity-duration) would behave differently.

Thank you for pointing these out. The present study focused on the most unfavorable wave type in terms of the flood mitigation with mangrove forest. The longer the wave period, the smaller the wave attenuates.

Therefore, the authors believe that investigating the diurnal tide is ideal for observing the fundamental function of the mangrove embankment system. The tidal damping for semi-diurnal tidal components, which are half the periods of the diurnal tides, appears to be more remarkable compared to the diurnal tide, when their amplitudes are the same. From the view point of Eco-DRR, it should be noted that the effectiveness of the mangrove embankment system will be automatically assured for tsunamis, storm surges, and high waves if it is effective against astronomical tides.

Also, some reorganization and text improvement should help to facilitate the reading and an overall understanding of the paper main contributions. Some suggestions are included in the specific comments.

Finally, it should be clarified (maybe in discussion) that the paper refers only to tide attenuation and the consolidation of the embankment over time but other factors, such as the diversity of species, are also critical for flood reduction and good ecological

performance. Diverse species increase friction (manning coefficient) in the mangrove forests and reduces flooding but are also a key component to becoming functional ecosystems, in the context of Eco-DRR. Many restoration cases in the past have failed to recognize the species diversity as a clear factor in restoring mangrove forests for flood protection.

Thank you so much for giving this great comment. We agree the importance of the role of diversified species in reducing floods. Some statement such as below will be added in "4. Discussion" of the revised manuscript.

**Diversity of species**: Although beyond the scope of this study, the diversity of mangrove species planted in the embankment should also be critical for creating functional and balanced ecosystems, which eventually contribute to flood mitigation by increasing the bottom friction in the mangrove forests. The magnitude of response to nutrients varies across mangrove species. Thus, within the intertidal zone, different species prefer different elevations, salinities and inundation frequencies.

Specific comments

Page 2 line 8-10 – Phrase unclear. line 10-12 – rephrase. Maybe: mangroves provide key ecosystem services such as: : : : Also, explain ecological resilience in this context.

The corrections will be made in the revised manuscript.

line 6. Move 'salt marshes have also : : :' after 'vegetation decreases: : : '

The corrections will be made in the revised manuscript.

line 21. Not all urban areas are suffering subsidence and sea level rise. Clarify and express better that: 'mangroves can help reduce risk in urban areas under threat from sea level rise and subsidence'

The corrections will be made in the revised manuscript.

line 29-31. See (Sasmito et al. 2015) for a discussion on mangroves and sea level rise. Page 3 Line 3 onwards. Clarify and define tides, surges and tsunamis. Line 13. Remove or describe sections at the end of the introduction Line 25-27. How is it affected by erosion of the waterfront? Clarify that erosion of the waterfront (for example from wave action) is not considered in the model.

The corrections will be made in the revised manuscript.

Line 29 onwards. While figure 1 describes the overall approach and design of the numerical experiment, the text in section 2.1 is described as conclusions rather than methods or hypothesis that the paper will test. It is suggested to rewrite this paragraph to better express what scenarios the experiment will address and compare.

The correction will be made in the revised manuscript.

Line 6. Describe differences between surges, tides and tsunamis and that only tides will be used in the paper.

The details on what type of coastal flood can be dealt with the proposed model will be clarified in the revised manuscript.

Page 5. Where is ue (eq 5) used in eq 6 and 7?

Equation 8 will be modified to explicitly indicate the parameter:

$$U(T) = \boxed{\int_0^1 \left(1 - \frac{u_e}{q_s}\right) dZ =} \ 1 - \sum_{n=0}^{\infty} \frac{2}{a_n^2} \exp(-a_n^2 T) \qquad (8)$$

Page 5 and 6. It is unclear where sediment deposition (later analyzed in the paper) appears in these equations. Is this through a linear sediment accretion rate (section 2.4)?

The details on this regard will be clarified in the revised manuscript.

Line 31. MSL : Mean Sea Level

The correction will be made in the revised manuscript.

Page 7. Section 2.5. Format or describe the case study.

Line 30. Consider removing.

The correction will be made in the revised manuscript.

Page 8. 1st paragraph. Only by soil consolidation?

Line 12. Because mangroves trap sediment: : : Line 24. Tide attenuation scenario is unclear. Line 26. Reduce the tidal energy? Unclear.

The correction will be made in the revised manuscript.

Page 9. Line 1-3. Outline better the 3 factors: plants height, water depth, embankment width and depth.

The correction will be made in the revised manuscript.

Line 4. Define vegetation growth rations. This variable is also used in subsequent figures but is unclear what it refers to.

The correction will be made in the revised manuscript. The present model defines the vegetation growth ratio to represent to what degree the mangrove plant grows. The ratio is assumed to linearly increase from 0% at the beginning and reaches 100% after 10 years.

Line 12-15. This paragraph is discussion material.

The paragraph will be moved to another section which would fit with this context.

Paragraph in line 20. How tidal amplitude influences the analysis? It is not defined in the paper that tide attenuation is only tested for a specific tidal amplitude and tide period. However, flood attenuation would depend on the wave amplitude and celerity (i.e.

period) in relation to the geometry of the embankment (depth and width), as seen in equations. In other words, for the same tide properties, different embankments geometries provide different attenuation factors, but for the same embankments, different tides would exhibit different behavior too. If sensitivity of the results to tides (and other flood events) is not explored, it should be properly acknowledged and described.

We consider that investigating the diurnal tide is sufficient for observing the fundamental function of the mangrove embankment system, which simultaneously experiences multiple external influences such as SLR, subsidence, and sediment accretion. Nevertheless, the discussion regarding different wave characteristics will be of great benefit to readers. Instead of giving additional simulated results for different tidal conditions, this appendix introduces some relevant mathematical expressions to discuss how a tide can behave differently under different conditions in terms of wave and platform geometries.

Line 30. The results presented confirm : : :

Page 10. Line 2. Use SLR consistently throughout Line 3. Tide amplitude and period too?

Line 19. Both potential and negative effects: Line 19. Paragraph also discussed in results. Consider consolidating.

The correction will be made in the revised manuscript.

Line 29. Erosion of embankments usually occurs from wave action at the forest front when the area is not properly sheltered from it.

We agree. The accelerated erosion at the embankment front will be stated in the revised manuscript.

Line 30. Diversity of species in the forest should also be consider to provide better frictional drag and functional ecological performance.

We agree. The diversity of species and its role in mitigating flood will be stated in the revised manuscript.

Page 11. Define and use MSL for mean sea level consistently throughout Line 4. Use static rather than stable. Line 6. Rewrite. For example: For example, climate patterns such as El Niño Southern Oscillation can increase sea level and these variations have been shown to larger than historic trends in sea level rise (e.g. Losada et al. 2013).

The correction will be made in the revised manuscript. Also, the suggested article will be cited to discuss the potential impact of ENSO.

The authors may consider adding a final discussion point on how these results could inform flood attenuation for other events that are not ocean tides.

Although this study focused on oceanic tides, the proposed methodology could also be useful to investigate the effectiveness of mangrove platforms against storm surges and tsunamis. In the revised manuscript, we will put more clarifications on the applicability of the model to the other coastal floods and also discuss limitations with that.

Line 16. Keep the mangrove surface level up with : : :

The correction will be made in the revised manuscript.

Line 20. These values of attenuation are specific to the tide amplitude and period. Technical comments on Figures Figure 1. It is confusing that ocean tides are here related to catastrophic flooding on urban areas (e.g. panel a). The sketch and text in the corresponding section should be more clear on what the sketches represent and what is the particular scope of the paper.

The scope of the paper will be more clarified by stating the type of coastal disasters concerned in the revised manuscript. Also, Figure 1 will be replaced by the revised figure, as shown below, in order not to give a false impression of astronomical tidal floods.

[Figure]

Figure 3. add the sedimentation rate. 1cm/yr?

The correction will be made in the revised manuscript.

Figure 4. Define "vegetation growth rates" and "growth ratio". Do they represent the same variable?

The correction will be made in the revised manuscript.

Figure 5. The 3D perspective is confusing. Suggestion: 3 panels, one for each water depth.

Indeed, there is some difficulty to realize the figure with three parameters. However, we consider it will also be useful to quickly see those interactions among the factors simultaneously. Thus, we would like to keep this figure as it was.